

# Effect of elevated temperature on membrane lipid saturation in Antarctic notothenioid fish

Vanita C. Malekar[1], James D. Morton[1], Richard N. Hider[1], Robert H. Cruickshank[2], Simon Hodge[3] and Victoria J. Metcalf[4]

[1] Department of Wine, Food and Molecular Biosciences, Faculty of Agriculture and Life Sciences, Lincoln University, Christchurch, New Zealand
[2] Department of Ecology, Faculty of Agriculture and Life Sciences, Lincoln University, Christchurch, New Zealand
[3] Department of Agricultural Sciences, Faculty of Agriculture and Life Sciences, Lincoln University, Christchurch, New Zealand
[4] Office of the Prime Minister's Chief Science Advisor, University of Auckland, Auckland, New Zealand

Corresponding author
Vanita C. Malekar,
Vanita.Malekar@lincolnuni.ac.nz

## ABSTRACT

Homeoviscous adaptation (HVA) is a key cellular response by which fish protect their membranes against thermal stress. We investigated evolutionary HVA (long time scale) in Antarctic and non-Antarctic fish. Membrane lipid composition was determined for four Perciformes fish: two closely related Antarctic notothenioid species (*Trematomus bernacchii* and *Pagothenia borchgrevinki*); a diversified related notothenioid Antarctic icefish (*Chionodraco hamatus*); and a New Zealand species (*Notolabrus celidotus*). The membrane lipid compositions were consistent across the three Antarctic species and these were significantly different from that of the New Zealand species. Furthermore, acclimatory HVA (short time periods with seasonal changes) was investigated to determine whether stenothermal Antarctic fish, which evolved in the cold, stable environment of the Southern Ocean, have lost the acclimatory capacity to modulate their membrane saturation states, making them vulnerable to anthropogenic global warming. We compared liver membrane lipid composition in two closely related Antarctic fish species acclimated at 0 °C (control temperature), 4 °C for a period of 14 days in *T. bernacchii* and 28 days for *P. borchgrevinki,* and 6 °C for 7 days in both species. Thermal acclimation at 4 °C did not result in changed membrane saturation states in either Antarctic species. Despite this, membrane functions were not compromised, as indicated by declining serum osmolality, implying positive compensation by enhanced hypo-osmoregulation. Increasing the temperature to 6 °C did not change the membrane lipids of *P. borchgrevinki.* However, in *T. bernacchii,* thermal acclimation at 6 °C resulted in an increase of membrane saturated fatty acids and a decline in unsaturated fatty acids. This is the first study to show a homeoviscous response to higher temperatures in an Antarctic fish, although for only one of the two species examined.

## INTRODUCTION

When the cell membranes of fish and other poikilothermic organisms are subjected to thermal change, modifications in membrane lipids and fluidity may occur in order to maintain membrane properties and functions. Altered membrane composition in response to lower or higher temperature, known as homeoviscous adaptation (HVA), is observed across all poikilotherms (*Hazel & Williams, 1990*). HVAs that occur over short periods during the lifetime of an individual are acclimatory adaptive changes; e.g., as observed in eurythermic temperate fish, which possess a broad thermal adaptable range. Acclimation to lower temperature results in increases in the proportion of unsaturated fatty acids in membranes, to allow optimal membrane fluidity to be maintained. This suggests a protective role of the homeoviscous response in short-term acclimation (*Skalli et al., 2006*; *Snyder, Schregel & Wei, 2012*). In contrast to these acclimatory adaptive changes observed in non-Antarctic fish, the HVA response in Antarctic fish can be a long-term evolutionary adaptive change in response to the low temperatures experienced in the Southern Ocean. Antarctic notothenioid fish, for example, display an evolutionary adaptive mechanism where their cell membranes possess an increased proportion of unsaturated fatty acids (*Hazel, 1995*).

Marine Antarctic ectotherms are stenothermal as they experience negligible seasonal variation ($-1.9$ °C to approximately 1.8 °C), resulting in limited ability to adapt to temperature variation (*Somero, 2010*), and increased vulnerability to climate change effects (*Aronson et al., 2011*). In temperate and tropical latitudes, marine ectotherms experience much greater seasonal variation in temperature, and are correspondingly more thermally tolerant or eurythermal (*Aronson et al., 2011*). Evolution of stenotherms in a 'stable ice bath' has involved many critical changes in the genome that facilitate life in extreme cold, such as losses of certain traits that are no longer required (*Pörtner, Peck & Somero, 2007*). Loss of heat shock response has been observed in Antarctic fish (*Hofmann et al., 2000*), resulting in extreme stenothermality due to an incapability to minimise damage to their protein pool caused by elevated temperatures (*Podrabsky, 2009*). HVA to the constant cold temperatures of the Southern Ocean is one of the key evolutionary adaptive changes in Antarctic notothenioid fish, but it is not known whether they have the capacity to change their membrane saturation states in response to warmer temperatures, as one trade-off cost for stenothermality may be reduced adaptive capacity.

Anthropogenic global warming (AGW) poses a threat to polar and especially stenothermal polar species, and there is a need to determine the impact of warmer temperatures on the acclimatory responses of these species, including cellular membrane remodelling. There is evidence that Antarctic fish may not exhibit an acclimatory HVA response to transient temperature changes, as unchanged membrane lipid saturation states were observed in Antarctic fish (*Trematomus bernacchii* (TB) and *T. newnesi*) acclimated to a temperature of 4 °C for five weeks (*Gonzalez-Cabrera et al., 1995*).

However, these fish showed positive compensation with an increase in the Na+/K (+)-ATPase activity and a decline in the serum osmolality, implying that membrane functions were not compromised in spite of the unmodified saturation states. Temperature is also a major determinant of membrane-cholesterol levels, with high membrane cholesterol observed in warm-acclimated marine copepods (*Hassett & Crockett, 2009*). Levels of cholesterol have been shown to increase at higher temperatures resulting in reduced membrane fluidity (*Crockett, 1998*). It is not known whether Antarctic fish share this adaptive membrane cholesterol change in response to increased temperature.

This study aimed to investigate both evolutionary and acclimatory HVA responses in Antarctic fish, and brings together existing evidence, along with new experimental data, to understand the evolutionary adaptive response associated with cold tolerance. Firstly, to investigate evolutionary adaptive HVA in cold-water fish, we established the normal lipid saturation profile of liver tissue from three Antarctic fish species collected in their normal physiological temperature for comparison with a non-Antarctic New Zealand fish species. More specifically, we compared liver membrane lipid profiles of two closely related Antarctic species, *Pagothenia borchgrevinki* (PB) and *T. bernacchii*, and a more distantly related icefish species *Chionodraco hamatus* (CH) of Antarctic notothenioid fish, as well as the non-Antarctic Perciformes species *Notolabrus celidotus* (NC). Icefish have evolved a suite of physiological adaptations to account for their loss of haemoglobin (*Kock, 2005*), following their diversification from the other Antarctic notothenioids, and we sought to determine whether CH had a different membrane lipid profile to the two closely related Antarctic fish species (PB and TB). Previous study indicates that erythrocyte membranes of icefish have fluidity consistent with those of TB, but with observed lipid differences (*Palmerini et al., 2009*). It is unknown whether the membrane lipid composition of other icefish tissues, especially liver, also differs from other notothenioids. Secondly, we investigated the acclimatory response of Antarctic fish to higher temperatures by examining whether thermal stress at 4 and 6 °C resulted in membrane restructuring in two Antarctic fish species (PB and TB), as indicated by altered membrane saturation states and cholesterol content. We hypothesised that membrane saturation, the major thermal adaptive mechanism, would occur only at reduced levels in Antarctic notothenioid fish as a response to elevated temperatures due their stenothermal nature, and thus make them vulnerable to the effects of AGW.

## MATERIALS AND METHODS

### Fish samples

The fish species used in the study are described in Table 1, and details of the fish harvest and husbandry are provided in Supporting Information 2. The field study comprising thermal acclimation experiments were conducted in the laboratory facilities at Scott Base, Antarctica approved by Antarctica New Zealand (K058—2007/2008). The procedures of fish handling were approved by the Animal Ethics Committee at the University of Canterbury (AEC 2006/2R and 2008/11R). Liver tissue from Antarctic notothenioid and non-Antarctic fish species sampled from their normal habitat were

**Table 1 Fish species sampled and collection location.**

| Fish species | Family | Location | Adaptation temperature (°C) |
|---|---|---|---|
| *Trematomus bernacchii* | Nototheniidae | McMurdo Sound, Antarctica* | −1 to 1.9 |
| *Pagothenia borchgrevinki* | Nototheniidae | McMurdo Sound, Antarctica* | −1 to 1.9 |
| *Chionodraco hamatus* | Channicthyidae | Terra Nova Bay, Antarctica | −1 to 1.9 |
| *Notolabrus celidotus* | Labridae | Kaikoura, New Zealand | 9–13 |

Note:
* These fish were used for establishment of membrane lipid profiles (pre-acclimation controls) and for thermal acclimation studies.

taken for the establishment of normal lipid profiles. PB and TB and CH were compared with the non-Antarctic fish NC, a common native New Zealand Perciformes species (*Ayling & Cox, 1982*). NC is non-migratory and has a broad thermal range (eurythermal), experiencing daily and seasonal variations in temperature (*Jones, 1984*). NC has been studied as a model temperate species in studies investigating mitochondrial functions under thermal stress (*Iftikar & Hickey, 2013*; *Iftikar et al., 2014*; *Iftikar et al., 2015*). NC has also been compared with Antarctic species PB in a physiological study, to assess the association of anaerobic performance with cold habitat (*Tuckey & Davison, 2004*). In this study, PB and TB samples comprised the pre-acclimation controls of the thermal acclimation experiment described, while sampling locations of CH and NC are provided in Table 1.

## Thermal acclimation experimental design

Following the pre-acclimation period of 15 days, five randomly chosen individuals of each fish species were euthanised before the thermal acclimation experiment started and their tissues were harvested as an initial control prior to temperature treatment. The remaining fish of each species were randomly selected and placed in either static or flow-through tanks (limitations of the aquaria facilities meant some treatments were in static tanks) and kept in groups of no more than 10 fish per tank. The capacity of each tank being 10 L. There were three treatment temperatures, which were −1 °C control treatment, 4 and 6 °C acclimation temperature treatments respectively. The control and 4 °C treatments comprised of three tanks, while the 6 °C acclimation temperature, had only one tank. The initial temperature of all the tanks was −1 °C at the time of fish transfer and, for the acclimation temperature treatments, the water temperature was then gradually increased from −1 to 4, or 6 °C. The experimental temperature regime consisted of stepwise increases in temperature to the target temperature over a 24 h period (except in the case of the 6 °C treatment, in which case slower acclimation over three days was employed). The tanks were maintained at the treatment temperature ±0.3 °C using two heat exchangers connected to a feeder tank that contained thermostatically-controlled water heater. Where possible, flow-through tanks were used, but by necessity, some of the heat treatments required the use of static tanks with oxygen bubblers. For static tanks, 25% of the tank capacity was replaced daily to avoid accumulation of waste products and decreases in oxygen concentration. A cohort of fish held at as close to

environmental temperature as possible was used as a control treatment; in which case fish remained at −1 °C for the duration of the experiment. Further, they were fed ad libitum twice weekly during the acclimation period. PB was acclimated for 28 days but TB species was only acclimated for 14 days primarily due to limitations in the Antarctic aquaria space and duration of the field season. A 24L: 0D photoperiod was maintained to model the summer conditions in McMurdo Sound.

## Sampling of tissue and plasma samples

Fish ($n = 5$) of each treatment, (including controls) were euthanised, blood samples collected and tissues harvested at 1, 2, 3, 7, 14 and (in the case of PB only) 28 days post-acclimation, for 6 °C thermal acclimation tissues were harvested after seven days post-acclimation for both the species PB and TB. Sampling procedures were performed at the Scott Base Wet Laboratory, with air temperature below 5 °C. Routine anaesthetic exposure via transfer to seawater containing MS-222 (ethyl m-amino benzoate methane-sulfonate) was performed. Fish were anaesthetised for 5 min in a 0.1 g/L solution of MS222 dissolved in sea water. Details of the individual fish were then recorded such as fish morphology, weight, length, sex, maturity stage, blood volume, parasite burden and tissues collected. Blood was collected from the caudal vein with a needled syringe (22 guage) and the fish was euthanised by severing the spinal cord. Fish were dissected using standard dissection procedures under sterile conditions and tissue samples were transferred into labelled vials and immediately frozen in liquid nitrogen. Tissues (liver, brain, heart, kidney, white muscle, red muscle, subcutaneous fat and adipose tissue) were rapidly removed, snap frozen in liquid nitrogen and stored at −80 °C for later biochemical and genetic analyses.

## Phospholipid fatty acid analysis

### Lipid extraction

The lipid extraction method followed that of *Folch, Lees & Sloane Stanley (1957)* but was modified as follows. Total lipids were extracted from 0.2 g frozen liver tissue. Samples were ground under liquid nitrogen and suspended in 6 ml of dichloromethane/methanol 2:1, 0.01% butylated hydroxytoluene (BHT). After sonication (W-225 from Watson Victor) for 5 min, 2 ml of 0.88% potassium chloride was added. The samples were vortexed for 2 min and centrifuged at 1,000$g$ for 5 min. The aqueous layer was re-extracted with 2 ml of dichloromethane/methanol 2:1, 0.01% BHT and the two organic layers combined and dried under nitrogen. Dried samples were then stored at 4 °C until fractionation.

### Lipid fractionation

The lipid fractionation method followed that of *Zelles (1997)*, but was modified as follows. Phospholipids were separated by re-suspending the total lipid extracts in chloroform and loading on to a solid phase extraction column (Biotage isolute SI 500 mg 6 ml SPE). The sample was allowed to stand for 2 min in the column and the lipids sequentially eluted with 5 ml of chloroform for elution of neutral lipids, 5 ml of acetone for glycolipids

and 5 ml of methanol for phospholipids. The phospholipid fraction was dried with nitrogen then stored at 4 °C before proceeding with methylation.

### Methylation

The tubes containing the evaporated samples were brought to room temperature and 1 ml of tetrahydrofuran: methanol (1:1v/v) was added, then vortexed for 30 s. 1 ml of 0.2M potassium hydroxide was added followed by 30 s vortex and incubation at 37 °C for 15 min. After incubation, 2 ml of hexane: chloroform (4:1) plus 0.3 ml of 1M acetic acid and 2 ml of deionised water were added and vortexed for 1 min followed by centrifugation at 1,000$g$ for 5 min. The top organic layer was transferred to a holding tube and 2 ml of hexane: chloroform (4:1) was added to the lower aqueous layer and vortexed for 1 min followed by centrifugation at 1,000$g$ for 5 min. The top organic layer was transferred to the holding tube containing the first organic fraction. The organic layer was evaporated under $N_2$ in a water bath at 37 °C. Hexane (50 μl) was added to the evaporated organic layer and this was then transferred to a 150 μl insert with a poly spring held in an amber vial for GC analysis.

### Gas chromatographic separation

Fatty acid methyl esters were analysed on a Shimadzu GC-2010 Gas Chromatograph (Shimadzu, Tokyo, Japan) fitted with a silica capillary column (Varian CP7420, 100 m, ID 0.25 mm, film thickness 0.25 μm, Serial # 6005241) and helium flow 0.96 ml/min. The split ratio was 15 to 1 and the injector temperature was 250 °C. The initial column temperature was 45 °C for 4 min, then ramped at 13 °C/min to 175 °C held for 27 min before another ramp of 4 °C/min to 215 °C. This temperature was held for 35 min before a final ramp 25 °C/min to 245 °C for 5 min. All GLC conditions were based on adapting the initial conditions indicated by *Lee & Tweed (2008)*. A flame ionisation detector was used at 310 °C and fatty acids were identified by comparison of retention times to standards (GLC 463; NuChek, Elysian, MN, USA). Known fatty acids are reported as a percentage of total fatty acids and fatty acids less than 1% were not reported.

## Membrane cholesterol analysis

Cholesterol was extracted with dichloromethane: methanol from 50 mg of liver tissue re-suspended in 1 ml of 2-methoxymethane, then stored at −80 °C (*Gonzalez, Odjélé & Weber, 2013*). The free cholesterol was measured using the cholesterol fluorometric assay 10007640 following the manufacturer's instructions (https://www.caymanchem.com/pdfs/, Kit item number 10007640) and read on a Fluorostar omega microplate reader (BMG Labtech, Offenburg, Germany).

## Plasma osmolality determination

### Collection and storage of plasma samples

Blood samples from the experimental Antarctic fish (TB, PB) were collected at Scott Base Wet Laboratory. The temperature of the Wet Laboratory was constantly below 5 °C. The experimental fish were anaesthetised for 5 min by administration of 0.1 g/L solution of MS222 (ethyl m-amino benzoate methane sulphonate) dissolved in sea water.

Blood samples were immediately drawn by cardiac puncture with a 25 gauge needle. Blood volume of 0.5–1.0 ml was collected into a tube containing anti-coagulant. The collected blood was centrifuged at 3,000$g$ for 2 min for the plasma separation. The resultant blood plasma was collected and snap frozen in liquid nitrogen and transported to New Zealand in an insulated container containing dry ice and then stored at $-80$ °C. Plasma from fish samples from both the species acclimated to 4 °C and the control temperature of 0 °C, and collected at all the time-points, were taken for osmolality analysis. The plasma samples were thawed to room temperature and 10 μl plasma aliquots were taken for osmolality determination. Osmolality was measured using a Wescor 5520 C vapour pressure osmometer, which was calibrated with standard solutions before the measurements.

## Calculations and statistics

All statistical analysis was performed using Minitab v17 software. Comparison of lipid profiles of the different species was performed using principal component analysis (PCA) based on a correlation matrix. The raw data consisted of a matrix of the percent contribution of each phospholipid fatty acid in each sample. The data were not transformed prior to analysis. One-way ANOVA followed by a Holm–Sidak post hoc test was performed to compare individual fatty acids among the four fish species.

Desaturase index (DSI) for Δ9-desaturase/Stearoyl-CoA desaturase (SCD) was calculated as the ratio of product to precursor of the individual fatty acids using the formula: C16:1n7/C16:0 and C18:1n9/C18:0 (*Cormier et al., 2014*). A total of two particular unsaturated fatty acids (C16:1n7 and C18:1n9) were used for DSI, as the ratio of (C16:1n7/C16:0) and (C18:1n9/C18:0) has been shown to correlate with SCD activity, degree of desaturation and membrane fluidity in a previous study (*Hsieh & Kuo, 2005*).

Two-way ANOVA was used to assess the effects of temperature (control 0 °C and acclimated 4 °C), acclimation time (days) and the interaction between temperature and time on plasma osmolality. A Holm–Sidak post hoc test was subsequently used to determine which treatments differed significantly. Remaining data analysis in the 4 and 6 °C thermal acclimation trials was performed using an unpaired Student's $t$-test.

To assess whether the data met the assumptions of ANOVA, an approximation of residuals to a normal distribution was established by visual inspection, and homogeneity of variances was confirmed using Bartlett's test. For one variable, the PB osmolality, variances were found to be significantly different among the treatment groups, even after transformation of the data (square root; log) was attempted. ANOVA is frequently considered robust against the assumption of equal variances, especially when sample sizes are approximately equal (*Ananda & Weerahandi, 1997*). Thus, we proceeded with the ANOVA in this case, but concede that the results should be treated with caution due to the reduced power of the test under these conditions.

# RESULTS AND DISCUSSION

## Novelty of the study and key results

This is the first study to show that higher temperature acclimation can induce a homeoviscous response in an Antarctic fish species; the response was dominated by

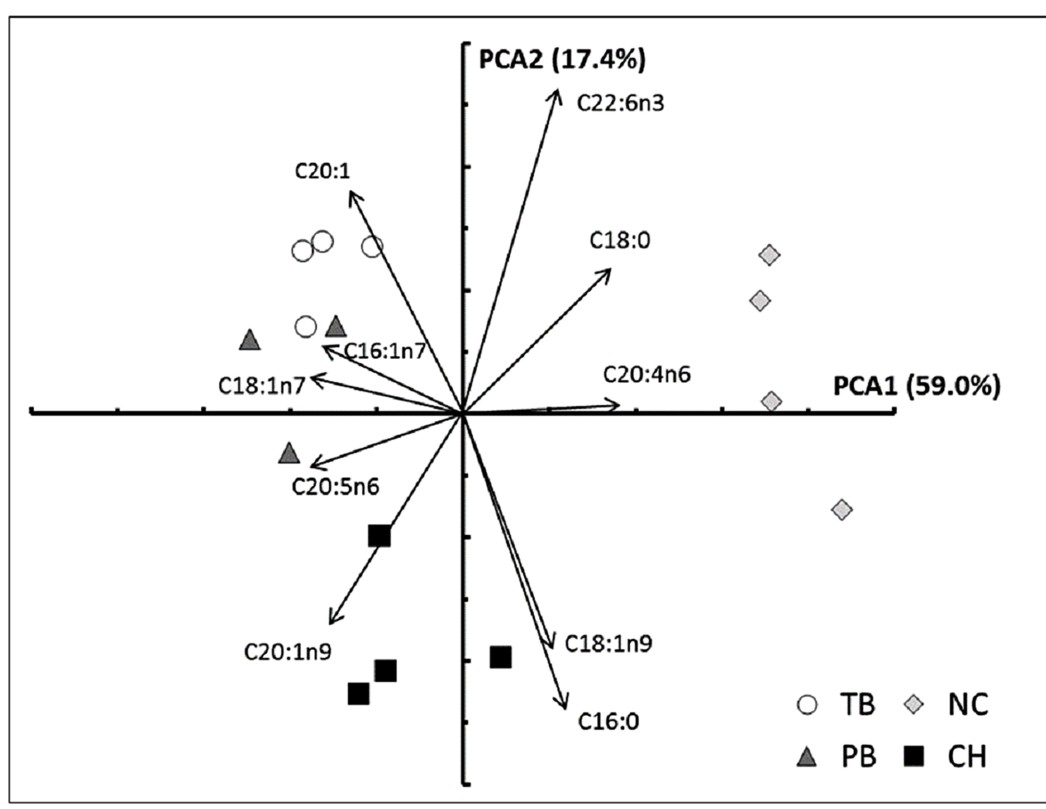

**Figure 1 PCA plot of the contribution of the phospholipid fatty acids to the principal components in liver tissue of the Antarctic species and non-Antarctic species.** Antarctic species *C. hamatus* (CH), *P. borchgrevinki* (PB) and *T. bernacchii* (TB) and the non-Antarctic species *N. celidotus* (NC).

changes in membrane unsaturation while membrane cholesterol remained unchanged. Our results also reveal that the presence of a homeoviscous response can vary depending on the Antarctic fish species. Thermal acclimation to 4 °C did not induce the HVA response in either of the Antarctic species TB or PB. However, an HVA response was induced in one of the Antarctic fish species, TB, when it was acclimated to a warming temperature of 6 °C. In addition, apart from palmitic acid which had similar levels in icefish and the non-Antarctic fish species, the membrane fatty acid composition of Antarctic fish species was found to differ from that of a non-Antarctic fish species at their respective typical environmental temperatures.

## Distinct phospholipid fatty acid composition of Antarctic species

The first two principal components of the PCA of the phospholipid profiles explained 76.4% of the variance in the data matrix. The PCA clearly separated the phospholipid profiles of the three Antarctic fish species (PB, TB and CH) from the samples obtained from the non-Antarctic species (NC) along PC1 (Fig. 1). The non-Antarctic fish NC was associated with high proportions of saturated fatty acids (SFAs; C18:0) and the polyunsaturated fatty acids (PUFA; C20:4n6), while the Antarctic species (all three) were associated with high monounsaturated fatty acids (MUFAs; C16:1n7, C18:1n7, C20:1n9

**Table 2 Fatty acid composition of phospholipids in liver of Antarctic (CH, PB, TB) and non-Antarctic fish (NC) expressed as % of total phospholipid fatty acids.**

|  | *N. celidotus* | *C. hamatus* | *P. borchgreviniki* | *T. bernacchii* |
|---|---|---|---|---|
| C16:0 | 20.05 ± 0.51[a] | 20.43 ± 0.69[a] | 13.16 ± 1.11[b] | 13.08 ± 0.52[b] |
| C18:0 | 12.64 ± 1.16[a] | 3.05 ± 0.56[b] | 4.16 ± 0.64[b] | 5.01 ± 0.29[b] |
| ΣSFA | 32.69 ± 1.45[a] | 23.47 ± 0.40[b] | 17.32 ± 1.55[c] | 18.09 ± 0.68[c] |
| C16:1n7 | 0.48 ± 0.48[b] | 3.30 ± 0.73[a] | 4.01 ± 0.41[a] | 4.97 ± 0.35[a] |
| C18:1n9 | 11.07 ± 1.70 | 10.51 ± 1.99 | 9.64 ± 1.12 | 7.42 ± 0.28 |
| C18:1n7 | 3.22 ± 0.50[b] | 7.69 ± 0.59[a] | 9.84 ± 0.74[a] | 9.11 ± 0.49[a] |
| C20:1n9 | nd | 5.04 ± 0.98 | 2.78 ± 0.14 | 3.62 ± 0.38 |
| C20:1* | nd | 0.87 ± 0.50 | 2.93 ± 1.83 | 4.24 ± 0.81 |
| ΣMUFA | 14.77 ± 1.39[b] | 27.40 ± 1.98[a] | 29.20 ± 0.38[a] | 31.99 ± 0.72[a] |
| C18:2n6 | 0.62 ± 0.62 | 1.17 ± 0.41 | nd | nd |
| C20:4n6 | 9.08 ± 0.50[a] | 4.97 ± 0.35[b] | 3.13 ± 0.24[b] | 4.79 ± 0.33[b] |
| C20:5n3 | 14.00 ± 0.75[b] | 19.83 ± 0.80[a] | 22.52 ± 1.92[a] | 19.63 ± 1.08[a] |
| C22:5n3 | 1.41 ± 0.47 | nd | nd | nd |
| C22:6n3 | 28.06 ± 1.53[a] | 20.84 ± 1.44[b] | 23.54 ± 1.36[ab] | 25.50 ± 0.32[ab] |
| ΣPUFA | 53.16 ± 1.96[a] | 46.80 ± 1.13[b] | 49.18 ± 0.50[ab] | 49.91 ± 0.73[ab] |

Notes:
Values are mean ± SEM ($n = 4$), except for *P. borchgrevinki* ($n = 3$).
nd, not detected.
Significant differences among the species for each particular fatty acid are indicated by different letter codes ($P < 0.05$).
* Unidentified MUFA.

and an unidentified MUFA, C20:1), and PUFA (C20:5n3) (Fig. 1; Table 2). Within the Antarctic species, the phospholipid profiles of closely-related species TB and PB were separated from those of the more distantly-related CH along PC2 (Fig. 1). CH was associated with relatively high proportions of the SFA C16:0 and the MUFAs C18:1n9 and C20:1n9, whereas PB and TB had higher levels of the MUFA C16:1n7 and the unidentified MUFA (C20:1).

Generally the eurythermal NC had a significantly higher total SFA and lower total MUFA when compared to the Antarctic fish species, but this distinction was not specific for the total PUFA (Table 2). Our results suggest that the Antarctic fish species membrane fatty acid profiles are relatively consistent and distinct when compared to the eurythermal species (NC) (Fig. 1; Table 2). Stenothermal fish species such as Antarctic fish exist in constant cold and have a narrow thermal adaptable range, and have been reported to have higher percentages of unsaturated fatty acids than temperate fish or eurythermal fish (*Logue et al., 2000*). Similarly liver microsomes of Antarctic fish *Disostichus mawsomi* had higher percentages of MUFA when compared to temperate fish such as trout and carp (*Römisch et al., 2003*). In our study it was the overall MUFA and some specific PUFA that were higher in Antarctic than the non-Antarctic fish species suggestive of a central role of MUFA than PUFA in cold adaptation for Antarctic fish, and this phenomenon is considered as part of adaptive homeoviscous response in the fish acquired over their evolutionary history (*Cossins, 1977*; *Hsieh & Kuo, 2005*; *Trueman et al., 2000*; *Williams & Hazel, 1995*). A key question of this study was to determine whether the recently diversified icefish (CH) differ in their membrane lipids when compared to the

other Antarctic fish species. This study shows that proportions of SFAs, primarily palmitic acid (C16:0), were similar in the Antarctic species CH and the non-Antarctic species NC; both of these species had significantly higher levels of C16:0 compared to the Antarctic species TB and PB (Table 2). Higher proportions of C16:0 in the membranes of the icefish liver could be one feature acquired after diversification from the other Antarctic species. A previous study on the erythrocyte membrane lipids of CH showed higher levels of unsaturated longer chain fatty acids such as C:20–C:22, while shorter chain fatty acids such as C:16 and C:18 became unsaturated in TB, with both species having consistent membrane fluidity (*Palmerini et al., 2009*). The icefish species could thus have evolved specific adaptations in liver membrane lipids, such as higher C16:0 levels in liver membranes, as shown in the present study, and unsaturation of longer chain fatty acids in erythrocyte cell membranes (*Palmerini et al., 2009*).

Palmitic acid was significantly lower for the two closely related Antarctic species (TB and PB) than the New Zealand species NC, and the other Antarctic species CH (Table 2), and also formed the major fraction of the total SFAs in the Antarctic fish species. Stearic acid (18:0) was significantly lower in all three Antarctic species and formed the minor fraction. Palmitic acid has a role in cold adaptation of membranes (*Farkas et al., 1994*) and may be the reason for the predominance of palmitic acid among the SFAs in our study (Table 2). These results align with a study comparing the phospholipid compositions of muscle tissue in 15 marine species from the southeast Brazilian coast and two species from East Antarctica, where palmitic acid comprised 54–63% of the total SFA content (*Visentainer et al., 2007*), and another study examining the total fatty acid content for all organs in two Antarctic species, *Notothenia coriiceps* and *N. rossii*, where palmitic acid represented 16–30% of the total FA content for all organs (*Magalhães et al., 2010*). Apart from high palmitic acid in Antarctic fish, increases in palmitic acid due to cold acclimation was observed in a study comparing two confamiliar species from different thermal habitats in the muscle of Antarctic eelpout, *Pachycara brachycephalum* in comparison to the temperate eelpout *Zoarces* (*Brodte et al., 2008*).

## Components of MUFA enhance membrane fluidity

All three Antarctic fish species were associated with high levels of MUFA associated with membrane fluidity, such as palmitoleic acid (C16:1n7), *cis*-vaccenic acid (C18:1n7), eicosenoate (C20:1n9) and total MUFA (Fig. 1; Table 2). Other studies have reported high *cis*-vaccenic acid in membranes of the Antarctic fish *Pleuragramma antarcticum* (*Mayzaud et al., 2011*), high latitude fish of the sub-Arctic (*Murzina et al., 2013*), and *Caenorhabditis elegans* worms exposed to cold (*Murray et al., 2007*). *Cis*-vaccenic acid has been shown to enhance glucose transport in adipocytes (*Pilch, Thompson & Czech, 1980*) and serotonin transport in endothelial cells (*Block & Edwards, 1987*). The conformation of unsaturated *cis*-vaccenic acid presents a possible structural advantage and has a potential role in maintaining membrane fluidity, which may be the reason for its selective incorporation in the membranes of Antarctic fish. Lower growth temperature has also been shown to increase the amount of *cis*-vaccenic acid in *Escherichia coli* and decrease the amount of palmitic acid incorporated in their membranes (*Marr & Ingraham, 1962*).

## EPA could offer additional roles other than membrane fluidity

Antarctic fish species had significantly lower levels of arachidonic acid (ARA, C20:4n6) and higher levels of eicosapentaenoic acid (EPA, C20:5n3) than non-Antarctic species (Fig. 1; Table 2). Levels of docosahexaenoic acid (DHA, C22:6n3) were not significantly different between Antarctic and non-Antarctic species (Table 2). Higher EPA proportions in the Antarctic fish species included in our study is in alignment with high EPA levels observed in muscle phospholipids of Antarctic fish from the Weddell and Lazarev Seas (*Hagen, Kattner & Friedrich, 2000*), in Antarctic silverfish, *P. antarcticum* (*Mayzaud et al., 2011*), in cold acclimated fresh water alewives (*Alosa pseudoharengus*) (*Snyder, Schregel & Wei, 2012*) and in cold acclimated *C. elegans* (*Murray et al., 2007*). Higher EPA in Antarctic species, and high EPA induced by cold acclimation in other species, suggest that EPA may play a role associated with cold tolerance, such as anti-inflammation or membrane stabilization. It has been suggested that DHA may possess a structural advantage over EPA in contributing to membrane fluidity due to the expanded molecular conformation of DHA (*Hashimoto, Hossain & Shido, 2006*). We did not see an increase in DHA and perhaps MUFA perform this role in Antarctic species. EPA, but not DHA, has been shown to be a potent anti-inflammatory agent, whereas ARA is highly pro-inflammatory (*Sears & Ricordi, 2011*; *Seki, Tani & Arita, 2009*). Hyper cholesteraemic rats, in whose membrane fluidity is reduced, have been shown to display increased membrane fluidity in their platelets when fed DHA but not when fed EPA (*Hashimoto, Hossain & Shido, 2006*). EPA may help in stabilization of hyper fluid membranes, as indicated by a study of the bacterium *Shewanella violacea* (*Usui et al., 2012*). EPA is one of the major (n-3) PUFAs present in the membranes of the Antarctic fish and contrary to other studies we do not observe correlation of DHA with membrane unsaturation, suggestive of modulation of particular fatty acids in HVA response. How these fatty acids (EPA, DHA and MUFA) contribute to fluidity and any other roles need further investigation in a larger range of fish species.

## Lack of distinction of membrane cholesterol between Antarctic fish and a New Zealand fish species

Membrane cholesterol was higher in the non-Antarctic New Zealand species NC than the Antarctic species PB, but not different to CH and TB (Fig. 2). In general, ectotherms adapted to lower temperature have shown to have reduced cholesterol levels primarily for maintenance of fluid state of membranes (*Crockett, 1998*). Contrary to the trend of a direct relationship with membrane cholesterol and habitat temperature, a higher percentage of cholesterol in muscle was observed in the higher Arctic fish species *Leptoclinus maculatus* in comparison to the related sub-Arctic species *Lumpenus fabricii* (*Murzina et al., 2013*). Currently there are limited data on the membrane cholesterol of Antarctic fish species. Our study showed cholesterol content varies with species, rather than the habitat temperature, a similar finding to those of *Palmerini et al. (2009)* where cholesterol in erythrocyte ghost membranes was highest in CH, followed by the non-Antarctic species *Anguilla anguilla,* and then lower in other Antarctic and non-Antarctic

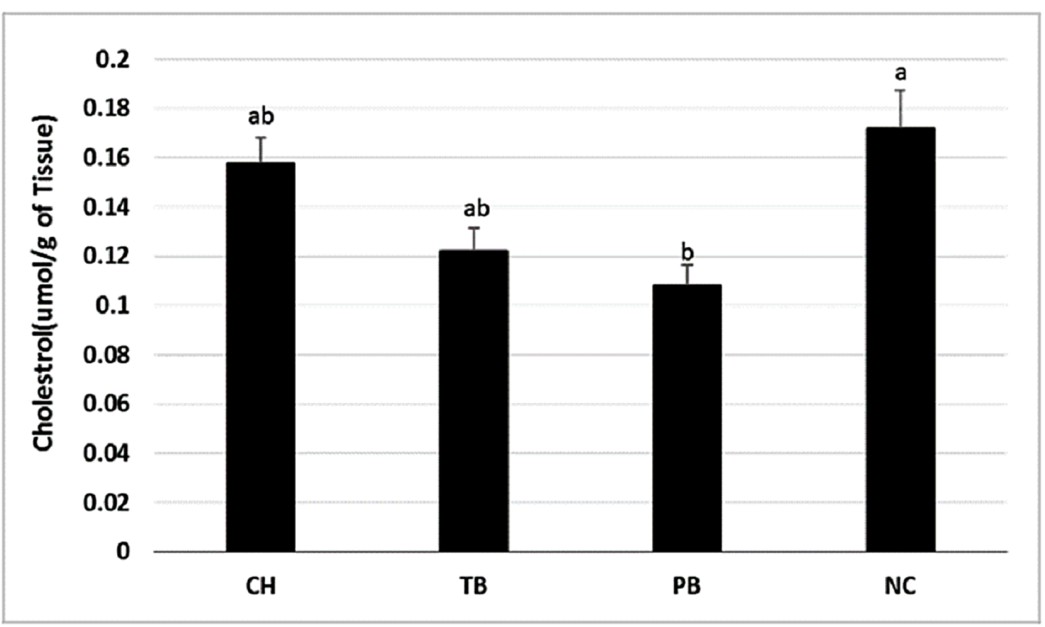

**Figure 2 Membrane cholesterol concentration in the livers of Antarctic species *C. hamatus* (CH), *P. borchgrevinki* (PB), *T. bernacchii* (TB) and non-Antarctic species *N. celidotus* (NC).** Values are mean ± SEM (*n* = 4). Significant effects among species are indicated by different letters (*P* < 0.05).

species. Thus, membrane cholesterol from further Antarctic species and from different tissues needs to be determined to establish its role in HVA.

## Lack of homeoviscous response in Antarctic species at 4 °C thermal acclimation

Thermal acclimation at 4 °C did not induce the major common cellular homeoviscous response in either the pelagic species (PB) or the benthic species (TB) after 28 or 14 days respectively (Table 3). There was no change in the DSI (C16:1n7/C16:0) and (C18:1n9/C18:0) in either species (Table 3). In TB, thermal acclimation changed the PUFA profile with a decrease in EPA (C20:5n3) levels and an increase in the amount of DHA (C22:6n3) (Table 3). As explained above, EPA levels may have a specific function in the extreme cold, perhaps in stabilizing membranes (*Usui et al., 2012*), or a protective role by reducing inflammation (*Sears & Ricordi, 2011*; *Seki, Tani & Arita, 2009*). The present findings of unchanged saturation states for PB and TB align with previous thermal acclimation experiments at 4 °C in the benthic Antarctic notothenioid species *T. bernacchii* and *T. newnesi*, where membrane unsaturation states were unchanged and there was no sign of an HVA response in the membranes of gills, kidneys, liver and muscle (*Gonzalez-Cabrera et al., 1995*). Similarly, mitochondrial membrane saturation states were also unchanged upon thermal acclimation and acidification, in the Antarctic species *N. rossii* acclimated at 7 °C and the sub-Antarctic species *Lepidonotothen squamifrons* acclimated at 9 °C (*Strobel et al., 2013*). Our findings have extended these observations to a cryopelagic species (PB), as well as confirming the lack of change in membrane saturation state in the benthic species TB.

**Table 3 Fatty acid composition of phospholipids in the liver of *T. bernacchii* (14 days acclimation) and *Pagothenia borchgrevinki* (28 days acclimation) acclimated at 0 °C and 4 °C.**

| | *Trematomus bernacchii* | | | *Pagothenia borchgrevinki* | | |
|---|---|---|---|---|---|---|
| | T0 | T4 | *P*-Value | T0 | T4 | *P*-Value |
| C16:0 | 16.01 ± 0.76 | 14.30 ± 0.53 | 0.13 | 13.79 ± 1.00 | 14.18 ± 0.34 | 0.74 |
| C18:0 | 6.49 ± 0.30 | 7.71 ± 0.47 | 0.08 | 5.33 ± 0.86 | 4.20 ± 0.37 | 0.29 |
| ΣSFA | 22.50 ± 1.00 | 22.01 ± 0.54 | 0.69 | 19.12 ± 1.5 | 18.38 ± 0.58 | 0.67 |
| C16:1n7 | 4.42 ± 0.78 | 3.09 ± 0.47 | 0.22 | 3.66 ± 0.38 | 4.17 ± 0.53 | 0.47 |
| C16:1 ≠ | 2.44 ± 0.87 | 2.68 ± 0.11 | 0.81 | 0.43 ± 0.43 | 1.81 ± 0.67 | 0.15 |
| C18:1n9 | 5.68 ± 0.26 | 5.98 ± 0.87 | 0.76 | 10.05 ± 0.68 | 11.27 ± 0.32 | 0.18 |
| C18:1n7 | 8.55 ± 0.23 | 9.03 ± 0.83 | 0.61 | 9.50 ± 0.50 | 9.69 ± 0.39 | 0.78 |
| C20:1n9 | 3.07 ± 0.27 | 3.26 ± 0.31 | 0.66 | 2.11 ± 0.20 | 2.05 ± 0.23 | 0.86 |
| C20:1 ≠ | 1.93 ± 1.20 | 2.83 ± 1.10 | 0.59 | 0.72 ± 0.72 | 1.17 ± 0.70 | 0.67 |
| ΣMUFA | 26.08 ± 1.20 | 26.87 ± 1.00 | 0.64 | 26.47 ± 0.83 | 30.15 ± 1.40 | 0.09 |
| C20:4n6 | 4.75 ± 0.73 | 4.68 ± 0.34 | 0.94 | 2.56 ± 0.10 | 3.49 ± 0.36 | 0.09 |
| C20:5n3 | 24.24 ± 1.10 | 19.05 ± 0.72 | 0.01* | 18.40 ± 1.30 | 17.64 ± 2.00 | 0.76 |
| C22:6n3 | 22.43 ± 0.64 | 27.38 ± 0.93 | 0.01* | 29.51 ± 2.90 | 30.35 ± 3.20 | 0.85 |
| ΣPUFA | 51.42 ± 2.00 | 51.11 ± 0.55 | 0.89 | 50.47 ± 2.30 | 51.47 ± 1.50 | 0.73 |
| DSI (desaturase index) | | | | | | |
| C16:1n7/C16:0 | 0.27 ± 0.04 | 0.21 ± 0.03 | 0.24 | 0.27 ± 0.02 | 0.30 ± 0.04 | 0.564 |
| C18:1n9/C18:0 | 0.88 ± 0.06 | 0.78 ± 0.11 | 0.46 | 2.11 ± 0.52 | 2.76 ± 0.31 | 0.341 |

**Notes:**
Values are mean ± SEM ($n = 4$) and expressed in % of total phospholipid fatty acids.
Significant effects of thermal acclimation are indicated by asterisks (*) ($P < 0.05$).
≠ Unidentified MUFA.

## Thermal acclimation has no effect on membrane cholesterol in the Antarctic species

Cholesterol is known to counter the effects of increased temperature on membrane lipids and an increase in cholesterol is often observed at high temperatures (*Crockett, 1998*). The structure of cholesterol mimics phospholipid structure and intercalates in the phospholipid membrane bilayer, resulting in an increase in membrane order and a reduction in membrane fluidity (*Crockett, 1998*). However, the membrane cholesterol in PB as well as TB was unaffected by thermal acclimation (Fig. 3; $P > 0.05$). This may be a tissue-specific effect as increased temperature resulted in a significant decline of cholesterol in the gill membranes of goldfish, but had no effect on the brain and liver cholesterol concentration (*Gonzalez, Odjélé & Weber, 2013*).

## Thermal acclimation results in a decline in plasma osmolality in both Antarctic species

Plasma osmolality gives an indication of the functioning of membranes. An inverse relationship exists between serum osmolality and water temperature. In an analysis of 11 teleost species, the serum concentration of Antarctic species was higher than the temperate species (*Dobbs & DeVries, 1975*). Fish inhabiting cold waters have high serum inorganic ion concentrations and these inorganic ions have been shown to have

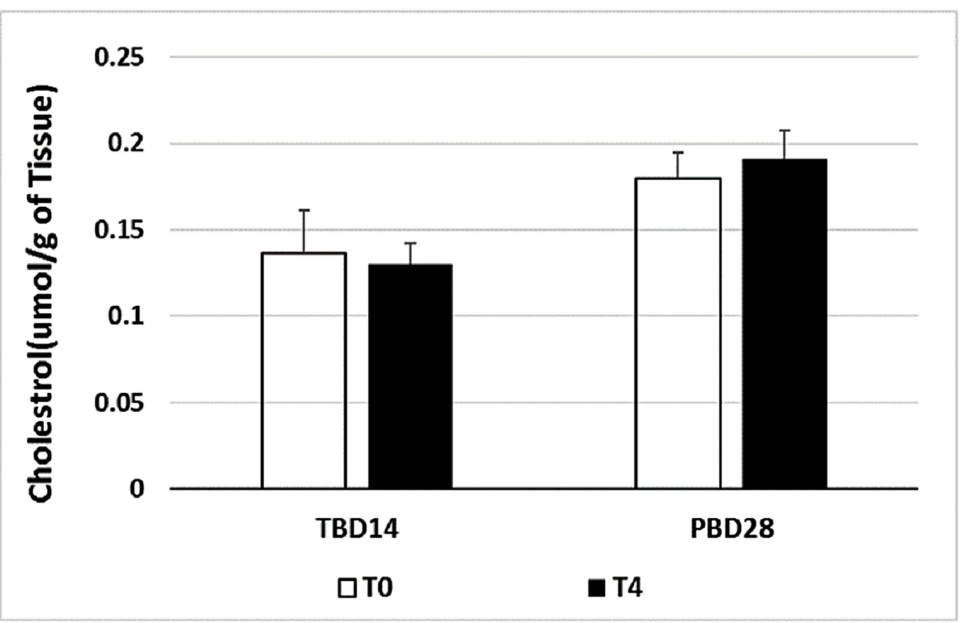

**Figure 3 Effect of thermal acclimation on membrane cholesterol concentration in the livers of *T.bernacchii* (TB) and *P. borchgrevinki* (PB).** Membrane cholesterol was determined 14 days after thermal acclimation in TB and 28 days in PB. Values are means ± SEM ($n = 4$) for control temperature (T0: 0 °C) and warm (T4: 4°C) acclimation.

protective roles in freezing avoidance by decreasing the melting point (*O'Grady & DeVries, 1982*). The plasma osmolality change over the 28 days of thermal acclimation at 4 °C in PB is presented in Fig. 4. Overall, irrespective of days of acclimation the osmolality at 4 °C was significantly lower in PB ($P < 0.01$), while a numerical but non-significant decline with temperature increase was observed for TB. The osmolality fell in both species after Day 3 of thermal acclimation and the reduction was significant at Day 7 ($P < 0.01$). Plasma osmolality in PB at 0 °C over the 28 days of acclimation remained unchanged ($P > 0.05$). The plasma osmolality showed a decreasing trend over the 14 day acclimation to 4 °C in TB, but this was not statistically significant (Fig. 4). In our study, thermal acclimation caused a decline in serum osmolality for PB. Other studies have also shown reduced osmolality upon thermal acclimation (*Gonzalez-Cabrera et al., 1995*; *Guynn, Dowd & Petzel, 2002*; *Hudson et al., 2008*; *Lowe & Davison, 2005*) which in some cases has been attributed to increased Na+/K(+)-ATPase activity (*Guynn, Dowd & Petzel, 2002*). The ability of these fish to control osmolality indicated that membranes were still functioning at 4 °C.

## Thermal acclimation at 6 °C results in an HVA response in *T. bernacchii*, but not in the pelagic species *P. borchgrevinki*

One of the key HVA responses is the change in the saturation states of membrane phospholipids (*Hazel, 1995*). TB exhibited an HVA response at 6 °C (Fig. 5), as shown by the increase in overall SFAs due to an increase in stearic acid, along with a decline in MUFA component eicosenoic acid (C20:1n9), total MUFAs and the PUFA component EPA (C20:5n3), while a significant increase in DHA (C22:6n3) was observed. SFAs reduce

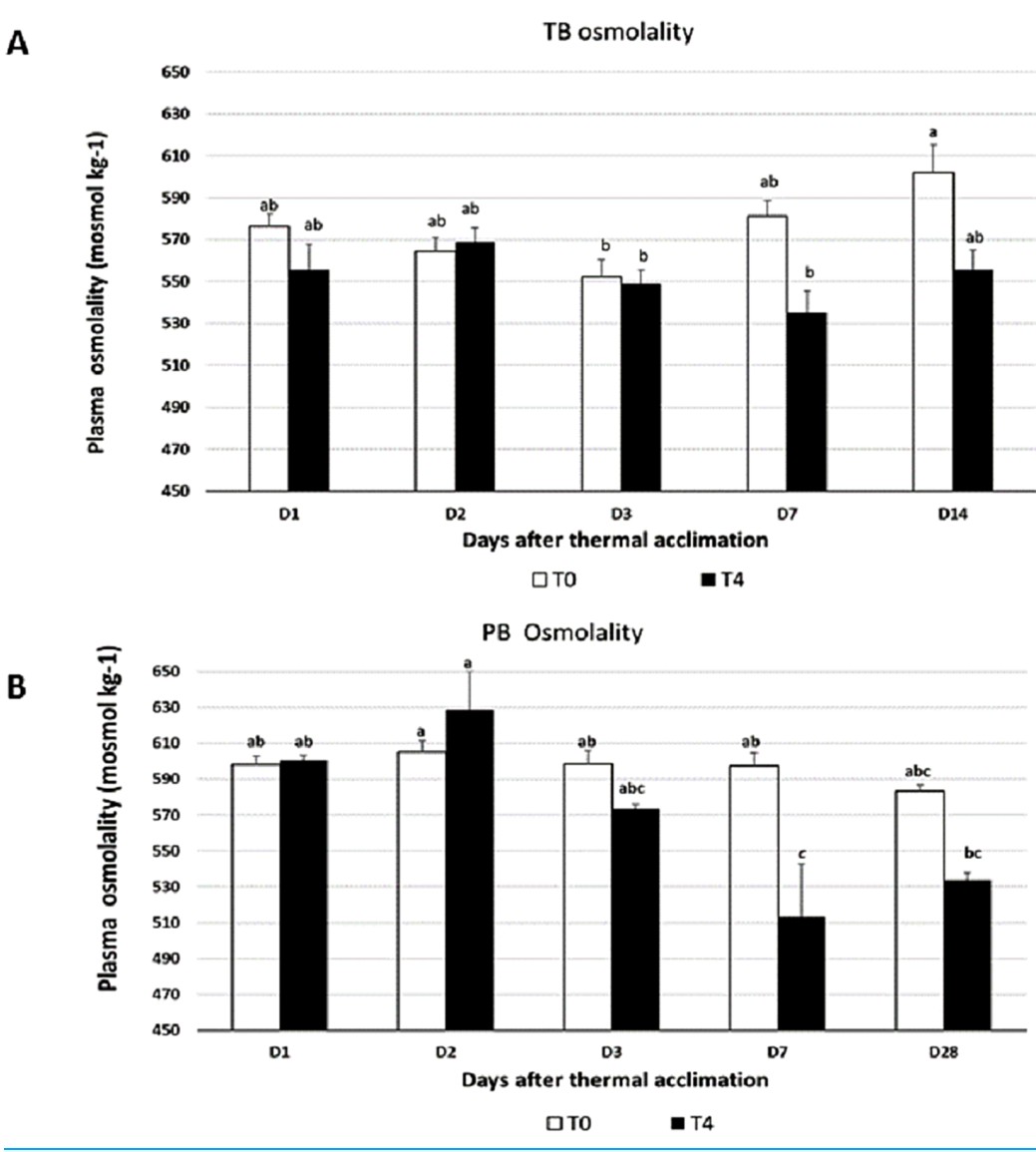

**Figure 4 Plasma osmolality determined at various time points.** (A) In *T. bernacchii* (TB). (B) In *P. borchgrevinki* (PB). Plasma osmolality was determined 14 days after thermal acclimation in TB (A) and 28 days in PB (B). Days after thermal acclimation D1, D2, D3, D7, D14 and D28 at 4 °C (T4) and the control temperature of 0 °C (T0). Values are ± SEM ($n = 4$). Significant effects of the interaction of thermal acclimation and days of acclimation are indicated by different letters.

membrane fluidity and offset the effects of increased temperature (*Hazel, 1995*). Previous studies of non-Antarctic fish species have shown that warm acclimation resulted in increased SFA and a decline in PUFAs *viz.,* EPA, DHA and ARA in brain phospholipids of *Dicentrarchus labrax* (*Skalli et al., 2006*), which has also been seen in fresh water alewives (*A. pseudoharengus*) (*Snyder, Schregel & Wei, 2012*). In yellow perch (*Perca flavescens*) warm acclimation resulted in decline of MUFA and PUFA in muscle phospholipids (*Fadhlaoui & Couture, 2016*). Although, the mechanism of HVA response upon warm acclimation is primarily dominated by a decrease in unsaturation, within this we observed

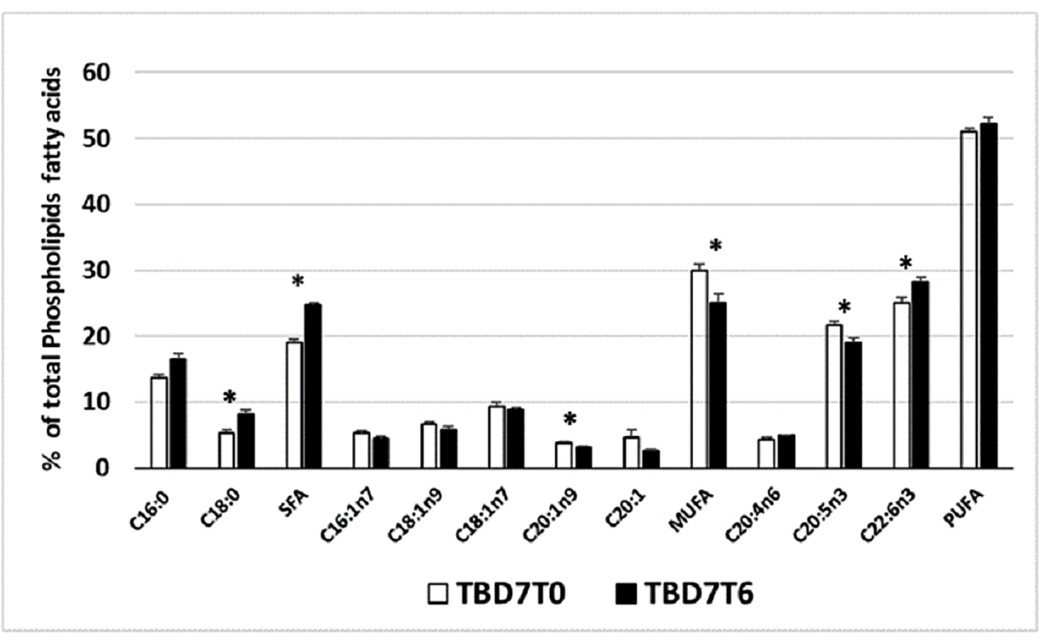

**Figure 5 Phospholipid profile of *T. bernacchii* (TB) in liver after 7 days (D7) of thermal acclimation at 6 °C.** Values are means ± SEM ($n = 4$) for control temperature (T0: 0 °C) and warm (T6: 6 °C) acclimation ($n = 3$). Significant effects of thermal acclimation are indicated by asterisks ($P < 0.05$).

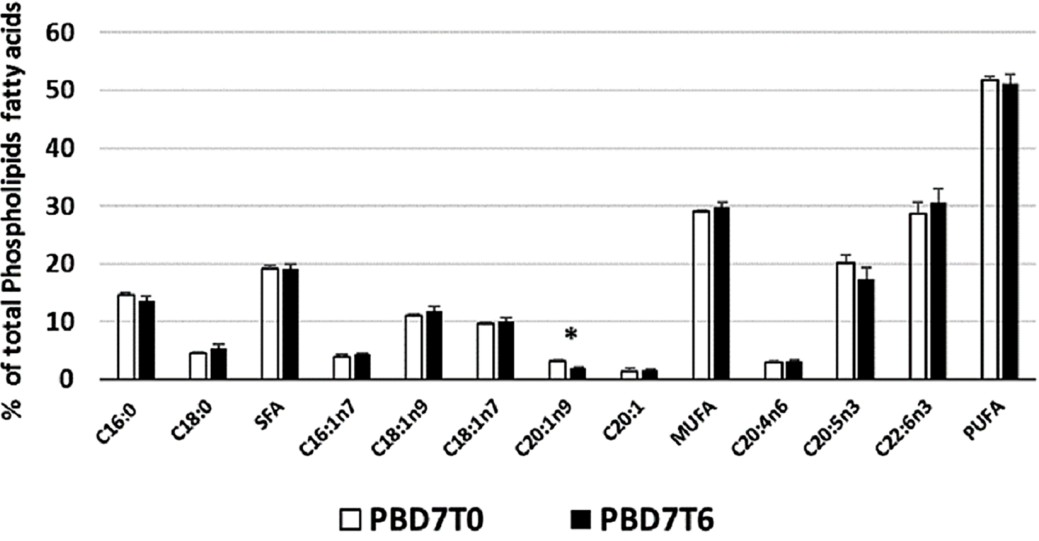

**Figure 6 Phospholipid profile of *P. borchgreviniki* (PB) in liver after seven days (D7) of thermal acclimation at 6 °C.** Values are means ± SEM ($n = 4$) for control temperature (T0: 0 °C) as well as warm (T6: 6 °C) acclimation. Significant effects of thermal acclimation are indicated by asterisks ($P < 0.05$).

an increase in DHA upon warm acclimation in TB at 4 °C (Table 3) and at 6 °C (Fig. 6), suggesting that particular fatty acids are modulated by temperature which could differ with tissue type and individual fish species. Tissue specific responses were also observed when warm acclimation induced an increase in DHA and palmitic acid in

goldfish liver, but not in brain, gill and muscle membrane lipids (*Gonzalez, Odjélé & Weber, 2013*) and also the role of DHA has been shown to vary among the eurythermal and stenothermal fish (*Brodte et al., 2008*). In eurythermal fish DHA is involved in cold acclimation as seen by increase in DHA of mitochondrial phospholipids with cold acclimation in rainbow trout (*Guderley et al., 1997*), similarly cold acclimation in carp resulted in DHA increase in liver phospholipids (*Farkas et al., 1980*). While in this study, increase in DHA in stenothermal Antarctic fish TB with temperature increase suggest DHA does not participate in cold adaptation. Similarly in the Antarctic fish *Pachychara bracycephalum,* high DHA correlated with temperature of highest growth in the muscle and liver tissue suggestive of a role in growth rather than with cold adaptation (*Brodte et al., 2008*). Thus our study supports the dual role of DHA depending on thermal environment of fishes.

In PB we found that 6 °C did not induce a significant HVA response (Fig. 6), although there was a decline in the MUFA component eicosenoic acid. Warm acclimation in both TB and PB caused a significant reduction of eicosenoic acid (Figs. 5 and 6). At their normal environmental temperature, these fish are found to have high proportions of eicosenoic acid in their membranes, as shown in analysis of the general phospholipid profile (Fig. 1), when compared to the New Zealand species, in which it was not detected. A similar role of eicosenoic acid in HVA response was observed in warm acclimated goldfish, with a decrease in the percent eicosenoic acid of brain and muscle phospholipids (*Gonzalez, Odjélé & Weber, 2013*). Apart from the reduction in eicoseonic acid, a major HVA response was not seen in PB. Other tissues may need to be analysed to confirm the apparent lack of a significant HVA response in PB. For example, the warm acclimation of *D. labrax* resulted in an HVA response in the brain, rather than the liver (*Skalli et al., 2006*). In another study, warm acclimation of the Antarctic species *N. rossii* at 7 °C and *L. squamifrons* at 9 °C did not result in an HVA response in mitochondrial membranes (*Strobel et al., 2013*). PB has a higher degree of thermal plasticity (*Franklin, Davison & Seebacher, 2007*) and higher upper lethal temperature compared with TB (*Somero & DeVries, 1967*). Thus, temperatures greater than 6 °C may be required to induce an HVA response in PB.

## Desaturase index correlates with membrane saturation state

In the present study the DSI (C16:1n7/C16:0) and (C18:1n9/C18:0) were shown to correlate with the saturation states of the membrane and DSI has been used as a surrogate for the measurement of SCD enzyme activity in membrane remodelling in response to temperature (*Fadhlaoui & Couture, 2016*). The enzyme SCD plays a key role in unsaturation of SFA by catalysing the synthesis of MUFA, primarily by the introduction of the first double bond between the C9 and C10 position of the fatty acid which results in increased membrane disorder and enhanced fluidity (*Paton & Ntambi, 2009*). The Antarctic species had a high DSI (C16:1n7/C16:0) compared to the non-Antarctic species, whereas this trend is not specific for the DSI (C18:1n9/C18:0) (Fig. 7). High DSI (C16:1n7/C16:0) in the Antarctic fish species could be attributed to an increase in the MUFA palmitoleic acid C16:1n7 reflecting higher desaturation of palmitic acid by SCD.

Peer

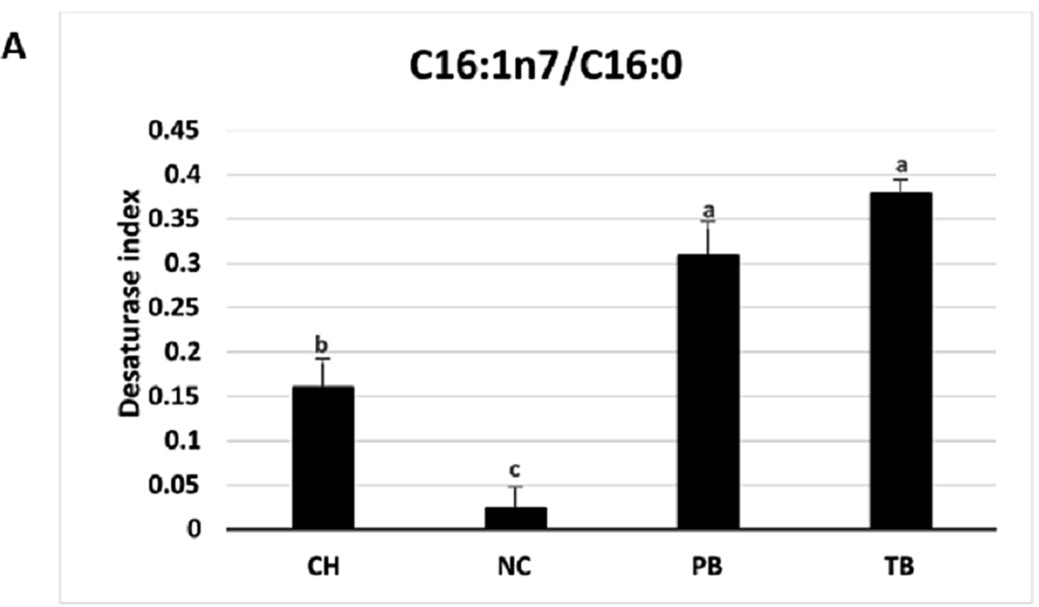

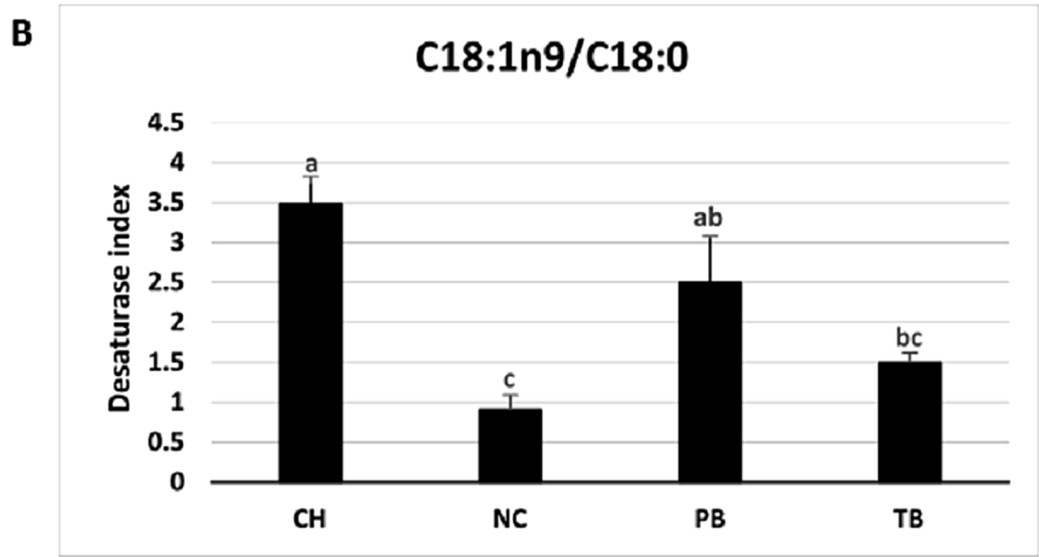

**Figure 7 Desaturase index in livers of Antarctic species *C. hamatus* (CH), *P. borchgrevinki* (PB), and *T. bernacchii* (TB) and the non-Antarctic species *N. celidotus* (NC).** (A) Desaturase index (C16:1n7/C16:0). (B) Desaturase index (C18:1n9/C18:0). Values are mean ± SEM ($n = 4$). Significant effects among species are indicated by different letters ($P < 0.05$).

Furthermore, in this study there was significant decline in DSI (C16:1n7/C16:0) upon thermal acclimation at 6 °C in *T. bernacchii* (Fig. 8). A positive correlation does exist with the DSI and membrane saturation states, as previously established in two fish species, milk fish and the grass carp when subjected to cold acclimation from 25 to 15 °C over 21 days (*Hsieh & Kuo, 2005*). Similarly higher DSI for SCD-18 (stearate desaturase) was observed in yellow perch (*P. flavescens*) acclimated at 9 °C than at 28 °C (*Fadhlaoui & Couture, 2016*). Furthermore our study supports this change specifically for the palmitate

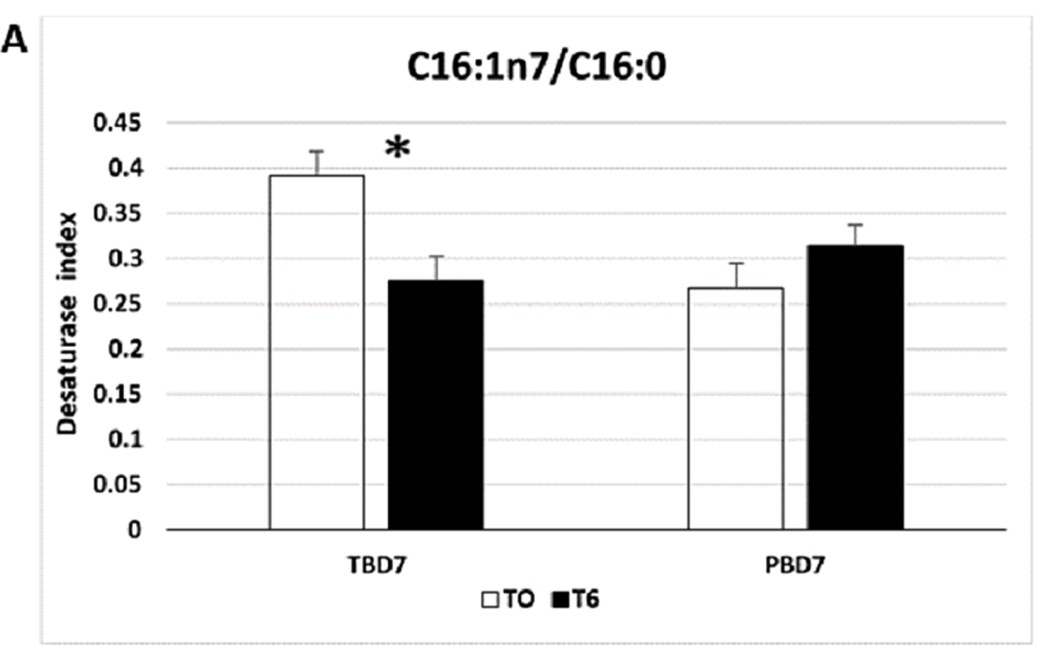

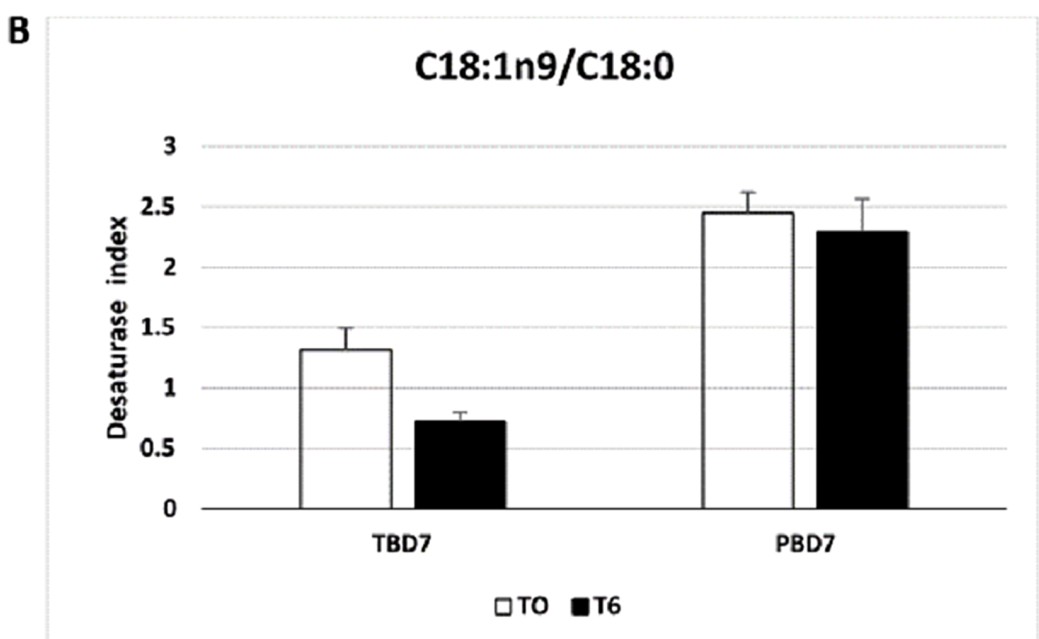

**Figure 8  Changes in the desaturase index in the livers of *P. borchgrevinki* (PB) and *T. bernacchii* (TB) acclimated at 6 °C for seven days.** (A) Desaturase index (C16:1n7/C16:0). (B) Desaturase index (C18:1n9/C18:0). Values are means ± SEM ($n = 4$) for control temperature (T0: 0 °C) as well as warm (T6: 6 °C) acclimation. Significant effects of thermal acclimation are indicated by asterisks ($P < 0.05$).

DSI (C16:1n7/C16:0) rather than for the stearate DSI (C18:1n9/C18:0) (Figs. 7 and 8). Hence the Antarctic species could display specificity for the palmitate desaturase activity for the HVA response. However, DSI provides limited information as it does not convey the complete picture of the lipid saturation, and data on the storage lipid dynamics is needed to establish the complete correlation of DSI. Future studies are needed

to assess the compartmentalization of DSI for the membrane lipid saturation, fatty acid synthesis, and chain elongation. A study on the compartmentalization of SCD1 activity in HepG2 cell lines have provided additional information on lipid pathways by determination of desaturation index in HepG2 cell lines using labelled stearate or palmitate (*Yee et al., 2008*).

## CONCLUSIONS AND PERSPECTIVES

This study has established a consistent membrane lipid profile across three notothenioid Antarctic species, in contrast to a varying membrane lipid composition between Antarctic species and a non-Antarctic New Zealand species. The Antarctic fish exhibit an evolutionary HVA response, as reflected by high levels of unsaturated fatty acids and selective dominance of *cis*-vaccenic acid and EPA in their membranes. This calls for further analysis of a wide range of fish species from different thermal habitats to decipher the specific roles of *cis*-vaccenic acid and EPA in cold adaptation. Previously undetermined is whether Antarctic fish can protect their membranes by exhibiting the acclimatory HVA response, which may make them less vulnerable to the effects of AGW. Our findings suggest that at 4 °C neither of the closely related Antarctic species exhibited any significant HVA response either with phospholipid unsaturation or with membrane cholesterol, but membrane-associated functions such as osmoregulation remain uncompromised. Furthermore, acclimatory HVA response of membrane unsaturation was detected at 6 °C in the liver of the benthic species TB while this response was lacking in liver membranes of the cryopelagic species PB. In the present study, HVA response was dominated by phospholipid unsaturation with no change in membrane cholesterol and the potential role of cholesterol in HVA response in Antarctic fish still remain unclear. Future studies especially at higher temperature acclimation as well as in other tissues are needed to determine the role of membrane cholesterol to HVA response in Antarctic fish. In conclusion, it appears that some Antarctic fish species can exhibit a limited HVA response to warming temperatures after a given acclimation period. However, this study has reinforced the need for further experimental work involving more species, over a wider range of acclimation temperatures and assaying multiple tissue types in order to ascertain the generality or specificity of acclimatory HVA responses in Antarctic fish.

## ACKNOWLEDGEMENTS

We thank Dr. Adrian Paterson for input to manuscript preparation.

### Funding

The field study for this experiment was supported by Antarctica New Zealand. Funding for biochemical analysis and paper writing was supported by Lincoln University New Zealand. The funders had no role in study design, data collection and analysis, decision to publish, or preparation of the manuscript.

## Grant Disclosures

The following grant information was disclosed by the authors:
Antarctica New Zealand.
Lincoln University New Zealand.

## Competing Interests

The authors declare that they have no competing interests.

## Author Contributions

- Vanita C. Malekar conceived and designed the experiments, performed the experiments, analysed the data, prepared figures and/or tables, authored or reviewed drafts of the paper, approved the final draft.
- James D. Morton conceived and designed the experiments, authored or reviewed drafts of the paper, approved the final draft.
- Richard N. Hider performed the experiments, approved the final draft.
- Robert H. Cruickshank project administration and supervision.
- Simon Hodge analysed the data, prepared figures and/or tables, authored or reviewed drafts of the paper, approved the final draft.
- Victoria J. Metcalf conceived and designed the experiments, performed the experiments, authored or reviewed drafts of the paper, approved the final draft.

## Animal Ethics

The following information was supplied relating to ethical approvals (i.e. approving body and any reference numbers):

The procedures of fish handling were approved by the Animal Ethics Committee at the University of Canterbury (AEC 2006/2R and 2008/11R).

## Field Study Permissions

The following information was supplied relating to field study approvals (i.e. approving body and any reference numbers):

The field study comprising thermal acclimation experiments were conducted in the laboratory facilities at Scott Base, Antarctica and approved by Antarctica New Zealand (K058—2007/2008).

## Data Availability

The raw data are provided as a Supplemental File.

## Supplemental Information

Supplemental information for this article can be found online at http://dx.doi.org/10.7717/peerj.4765#supplemental-information.

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
