# Peer review of "Effect of elevated temperature on membrane lipid saturation in Antarctic notothenioid fish"

_PeerJ, doi:10.7717/peerj.4765_

## Round 0.1 · original submission · Minor Revisions

I would suggest to include more information regarding the experimental design, particularly how fish were housed and the thermal challenges administered.

Reviewer 1 ·

Basic reporting

The article is well-written and the literature review is thorough and informative, especially in the Results and Discussion section. Raw data are supplied.

The authors might consider using different notation for fatty acids. Instead of using the format “C16:1c9”, the smaller “c” could be replaced with “Δ”. In physiological papers, it more common to use “C16:1n9” or C16:1ω9”.

Experimental design

Sample sizes are small and replication of experimental trials was not performed. However, given the difficulties in working with Antarctic fishes, this is understandable. The physiology of Antarctic fishes is not well-understood, so a major strength of this paper is providing new information on membrane fatty acid composition, membrane cholesterol, and changes during thermal acclimation.

Inclusion of the non-Antarctic, New Zealand fish species (N. celidotus) in the study needs to be justified. This species is in a different family (Labridae) than the Antarctic species so it is distantly related, and moreover, this species occurs in habitats that are considerably warmer. Phylogenetic and environmental effects are confounded, making it difficult to make meaningful comparisons between this species and the Antarctic fishes with respect to membrane fatty acid composition. I would consider removing it from the analysis.

Validity of the findings

In general, the analysis is appropriate and conclusions are reasonable and well-supported by citing appropriate previous work. The only significant question I have is the use of the desaturase index. Although the desaturase index may correlate with membrane saturation state, it does not necessarily reflect a cause and effect relationship. The decrease in the C16:1/C16:0 index reported by the authors could be due to a decrease in desaturase activity, but it could also be due to changes in mobilization and incorporation rates of specific fatty acids from storage lipids. If changes in desaturase are involved, then the situation is complex since the response appeared to occur with respect to C16:0 but not C18:0. Also, changes in C16:0 and C16:1 individually in response to thermal acclimation were not significant in TB according to figure 5, so my confidence in the actual significance of the desaturase index in this case is not particularly strong. Overall, I do not believe that the desaturase index contributes much to the overall paper since no direct enzymatic data or information regarding storage lipid dynamics are provided, so removing it from the paper should be considered.

A minor point: it is common with fatty acid percentage data to use arcsine transformations to meet assumptions of normality and homogeneity of variances for ANOVA. The authors state that untransformed data were used, so it should be clear that assumptions for the ANOVA were met without transformation of the data.

Additional comments

Overall the paper well-written and is a valuable contribution to our understanding of the physiology of Antarctic fishes. The two main issues that I would address are:

1. consider removing the data and comparisons involving the non-Antarctic fish since phylogenetic and environmental effects are confounded

2. consider removing the desaturase index since alternate explanations are available and no direct evidence for changes in enzyme activity are provided

Reviewer 2 ·

Basic reporting

The paper is written in clear English, is relatively easy to follow along, and is organized well.

Experimental design

I would appreciate more information on the housing and experimental setup being added to the manuscript rather than relying on the supplemental information (Expanding lines 116 - 121 to include numbers, time frame of acclimation to water temps, and how water temps were raised).

Validity of the findings

I believe the findings are relatively sound and are analyzed appropriately.
I would suggest removing the information in lines 358-363 trying to explain the day 2 spike in osmolality. Instead I would just focus on the general decrease in osmolality seen over time post acclimation.

Additional comments

Serveral citations are bit messed up, possibly due to some citation software being used. (lines 124,315).

There is an "unknown MUFA", if you know it is a MUFA then you probably know the carbon chain length but not the location of the double bone. It would be helpful to report this as 18:1 or 20:1 or whatever it is.

I would also remove line 75- 76 and combine lines 76 - 79 into the previous paragraph to wrap up changes that occur to membranes when water temperature rises.

For the PCA, consider using different symbol shapes for the non-antarctic fish, icefish, and antarctic fish to help differentiate them more than the gray scale color alone.

---

## Round 0.2 · accepted · Accept

We thank you for your patience and we hope that this new and improved publication structure will encourage you to submit your latest research results again to PeerJ

#